# Temporal Trends and Future Projections of Accumulated Temperature Changes in China

Xuan Li [1,2], Qian Yang [1], Lun Bao [2], Guangshuai Li [2], Jiaxin Yu [2], Xinyue Chang [2], Xiaohong Gao [2] and Lingxue Yu [2,*]

1    School of Geomatics and Prospecting Engineering, Jilin Jianzhu University, Changchun 130118, China
2    Remote Sensing and Geographic Information Research Center, Northeast Institute of Geography and Agroecology, Chinese Academy of Sciences, Changchun 130102, China
*    Correspondence: yulingxue@iga.ac.cn

**Abstract:** The Fifth IPCC Assessment Report indicates that climate change will affect crop growth and threaten the stability of food systems. Accumulated temperature, which is closely related to vegetation phenology and cropping systems, is an important indicator of heat in a region. Studying the history and future accumulated temperature changes can provide scientific reference for the change of crop phenology and cropping system, which is important for the improvement of grain production in China. Based on the MK trend test, MK abrupt change test and interpretable machine learning model, this study analyzes the spatial and temporal variation of accumulated temperature in China from 1979 to 2018, predicts its future variation based on CMIP6, and investigates the dominant influencing factors among different agricultural regions. The study found that (1) the accumulated temperature belt shows a northward shift and retreat trend toward higher altitudes, and the area of the high accumulated temperature belt increases year by year, leading to the narrowing of the area of the low accumulated temperature belt year by year, and the trend remains unchanged under the future scenario; meanwhile, the northward shift trend of the accumulated temperature belt is greatly mitigated and curbed under the SSP126 scenario. (2) The changes of accumulated temperature belt are mainly influenced by the increase of accumulated temperature duration days, and secondarily by the increase of temperature. The contribution brought by the first day of accumulated temperature from 1979 to 2018 is greater than that brought by the last day of accumulated temperature, while in the future scenario, on the contrary, changes in vegetation phenology delay should be given more attention.

**Keywords:** accumulated temperature; spatial and temporal variation; interpretable machine learning; main influence factor; CMIP6

## 1. Introduction

The Fifth Assessment Report of the Intergovernmental Panel on Climate Change (IPCC) indicates that by the end of this century, the global average temperature will increase by 0.3–4.5 °C compared to the end of the previous century. Global warming will inevitably lead to an increase in the frequency of extreme natural disasters [1]. Meanwhile, under the current high emissions scenario, human society will be more challenged than ever to cope with changes in global ecosystems, especially vegetation cover [2–4]. In addition, climate change will affect the food supply side, threatening the stability of food systems and causing unavoidable impacts on human socioeconomic and ecosystem balances [5–7].

The accumulated temperature (AT), as the sum of the average daily temperature in a region over a period of time, is greatly affected by the increase in global average temperature. The limiting temperature of 10 °C AT is an important indicator of the thermal status of a region and has an extremely important impact on crop growth in high latitudes while playing a key role in the cropping system transition in different growing areas [8]. In

addition, it has been shown that the increase in AT will lead to changes in crop yield and cropping structure [9], while the first and last days of AT are also closely related to changes in crop phenology changes [9]. Therefore, it is important to study the spatial and temporal variation of AT and its future trend and find out the main factors influencing AT variation for the adjustment of cropping structure, the cultivation of new crops as well as prevention of crop disasters.

The conclusions of the analysis of the spatial and temporal variation of AT are relatively uniform, while the conclusions of the analysis of the dominant factors of AT variation are more diverse. In China, the overall AT and regional AT show an increasing trend in time, the area of the low AT belt shows a decreasing trend, and the area of the high AT belt shows an increasing trend [10]. Spatially, AT is generally high in the south and low in the north, which is inversely proportional to the altitude [8]. From the perspective of the dominant factors of AT change, Dai S et al. analyzed the correlation coefficient between AT and mean annual temperature to conclude that the increase in AT is significantly linear with the increase in temperature (TMP) [11]. Zhao H et al. analyzed the correlation between the first day of accumulated temperature (AFD), the last of accumulated temperature (ALD), accumulated temperature duration days (ADD) and AT to find that the first or last day of AT has a significant influence on the change of AT [12]. In addition, Kong F et al. believed that the fluctuation of AT has a positive correlation with altitude and latitude [8]. However, China is a great agricultural country with a vast territory, a wide variety of crops, and diverse climatic types. Previous scholars have studied more from the perspective of local regions and geographic divisions but not much from the perspective of agricultural areas [12–16].

In terms of spatial interpolation methods and on-site data, common algorithms include inverse distance weight interpolation [17], multiple linear regression interpolation [18], and ANUSPLIN statistical interpolation [15]. On-grid data, common interpolation methods include bilinear interpolation [19] and statistical downscaling based on terrain correction [20]. From the analysis methods of spatial and temporal variation of AT, the common algorithms are the climate tendency method [21], intergenerational analysis [11], anomaly analysis [11], MK trend and the abrupt change test [9,17]. In terms of methods for analyzing the dominant factors of AT change, the commonly used methods include the correlation coefficient method [12]. It has been shown that the spatial characteristics obtained by interpolation extrapolation using grid point data are more precise than those obtained by interpolation of station data [10]. For temperature data, due to its close relationship with elevation, the statistical downscaling method based on terrain correction is more accurate than that obtained by bilinear interpolation when the terrain is complex [20]. The MK trend and abrupt change test are not subject to a certain distribution because they do not require the sample to follow a certain distribution and are not disturbed by a few outliers, so they are widely used in time-series data analysis [22]. Compared with the correlation coefficient method, interpretable machine learning can be interpreted simultaneously at both micro and macro levels, resulting in a better analysis of the contribution of influencing factors to the target variable [23–25].

Therefore, in this study, the statistical downscaling method based on terrain correction was selected to spatially interpolate the coarse CMIP6 grid point data to obtain higher resolution spatial grid data, MK trend and abrupt change test to analyze the spatial and temporal variation of AT, and interpretable machine learning to analyze the factors influencing AT. The main objectives of this study are: (1) To analyze the spatial and temporal variation of AT at 10 °C from the perspective of nine agricultural regions in China, and further to explore the influencing factors of AT in different agricultural regions. (2) To study the spatial and temporal variation of AT under different shared socioeconomic pathways, explore the evolution of AT zones under different shared socioeconomic pathways, and seek the key factors influencing AT change.

## 2. Data Sources and Methods

### 2.1. Overview of the Study Area

China has a large area of undulating terrain, complex topography, diverse climate types, and a wide variety of food crops. According to the Comprehensive Agricultural Zoning of China [17] prepared by the National Agricultural Zoning Committee, the country is divided into nine major agricultural zones: the Northeast China Plain (NEP), the Northern Arid and Semiarid region (NAS), the Huang-Huai-Hai Plain (3HP), the Loess Plateau (LP), the Qinghai-Tibet Plateau (QTP), the Middle-Lower Yangtze Plain (MYP), the Sichuan Basin and Surrounding regions (SBS), the Yunnan-Guizhou Plateau (YGP) and Southern China (SC) (Figure 1). Among them, NAS and LP's main food crops are maize, NEP's main food crops are maize and rice, 3HP's main food crops are maize and wheat. The main food crops of SBS, MYP, YGP, and SC are rice; QTP's main crops are highland barley (grown in Tibet and Qinghai) [26].

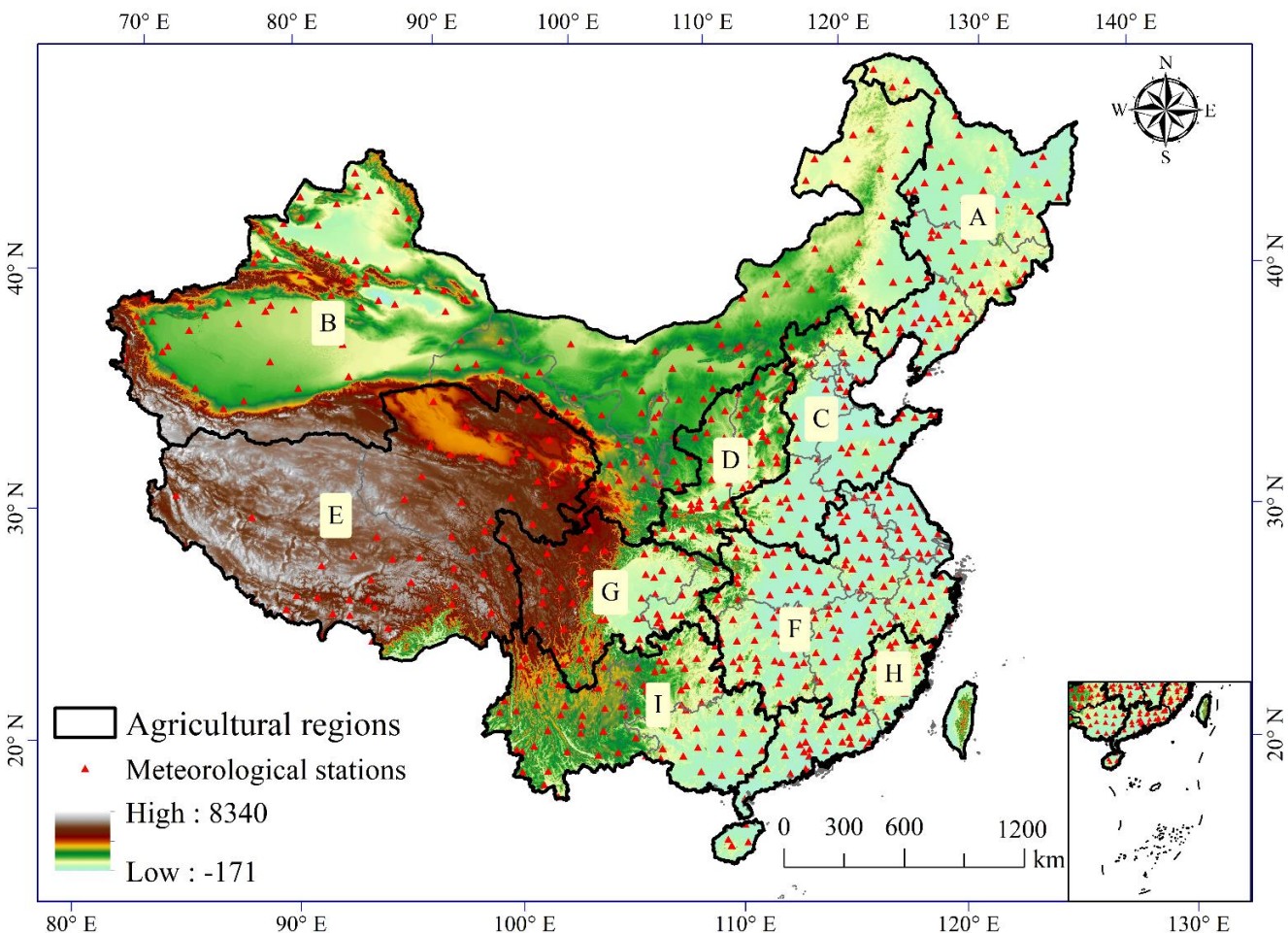

**Figure 1.** Nine major agricultural regions in China and the distribution of meteorological stations. The (**A–I**) are NEP, NAS, 3HP, LP, QTP, MYP, SBS, SC and YGP. The **red triangle** is the national meteorological station.

To facilitate the study of the spatial and temporal variation of ≥10 °C AT, the AT belt was divided according to the Chinese agricultural cropping zones for ≥10 °C AT. Namely, the tropical zone (>8000 °C × d) with three crops in one year, the subtropical zone (4500~8000 °C × d) with two to three crops in one year, the warm temperate zone (3400~4500 °C × d) with two crops in one year, the middle temperate zone (1600~3400 °C × d) and the cold temperate zone (<1600 °C × d) with one crop in one year.

*2.2. Data Sources*

2.2.1. China Meteorological Forcing Dataset (1979–2018)

The historical temperature data were obtained using the China Meteorological Forcing Dataset (CMFD), which consists of seven elements: surface air temperature, surface air pressure, surface air specific humidity, surface full wind speed, ground downward short-wave radiation, ground downward long-wave radiation, and ground precipitation rate. The data are in NETCDF format with a temporal resolution of 3 h and a spatial resolution of 0.1°, which can provide driving data for the simulation of land surface processes in the China region [27]. The dataset is produced using the internationally available Princeton reanalysis data, GLDAS data, GEWEX-SRB radiation data, and TRMM precipitation data as background fields and incorporating routine meteorological observations from the China Meteorological Administration (CMA). The original information was obtained from the CMA observations (observed near-surface meteorological data at about 700 weather stations in China), reanalysis information, and satellite remote sensing data. The values of non-physical ranges have been removed, and ANUSPLIN statistical interpolation is used. The accuracy is between the Meteorological Bureau observation data and satellite remote sensing data and better than the accuracy of the internationally available reanalysis data [28,29]. In this paper, the data of daily values of near-surface mean temperature from 1 January 1979 to 31 December 2018 were selected.

2.2.2. The ScenarioMIP Dataset

The future scenario data were used for four different scenarios from the Scenario Model Intercomparison Project (SSP1-2.6, SSP2-4.5, SSP3-7.0, and SSP5-8.5) of the Sixth International Coupled Model Comparison Program (CMIP6) [30,31]. In this paper, the ScenarioMIP data were selected to validate the model simulation capability with the data intersection part of the CMFD dataset, i.e., the temperature data from 1 January 2015 to 31 December 2018. The ScenarioMIP data and the CMFD data were interpolated to 800 meteorological stations nationwide, respectively, and 25 sets of series were obtained by calculating the average of the daily near-surface average temperature at each station from 1 January 2015 to 31 December 2018, of which one set is the daily average temperature of the CMFD dataset, i.e., the historical value, and 20 groups are the simulated mean daily average temperature values for each model under different scenarios, and four groups are the mean values of the simulated mean daily average temperature values for each model under each scenario. Then, the spatial Pearson correlation coefficients *P* and the ratio of the standard deviation of each model to the standard deviation of the historical values *σ* were calculated for the historical values of all stations with respect to the simulated values of other models and the average simulated values of multiple models, while a quantitative evaluation index *S* was introduced [32,33].

$$S = \frac{4(1+P)^4}{\left(\sigma + \frac{1}{\sigma}\right)^2 (1+P_{max})^4} \tag{1}$$

where *P* is the spatial Pearson correlation coefficient for each group corresponding to the historical values, $P_{max}$ is the maximum value of the spatial Pearson correlation coefficient for each group corresponding to the historical values, and *σ* is the standard deviation ratio. The calculation results are shown in the ScenarioMIP prediction and historical evaluation table (Table S1). According to the results in the table, the model with the correlation coefficient, standard deviation ratio, and S-value closest to 1 under different scenarios was selected for future scenario analysis, and the final data selected were the predicted values of the EC-Earth model under four scenarios.

*2.3. Methodology*

In this paper, the five-day rolling average method was used to calculate the AT at 10 °C, AFD, ALD, and ADD. The MK trend test was used to calculate the slope of each

grid point to study the spatial distribution of the AT tendency, whereby the grid points that did not pass the significance test at the 95% confidence level were excluded [22]. The area of the AT belt was counted, and the northward trend of the AT belt was analyzed using the least squares regression [10], and interpretable machine learning were used to determine the main influences on AT. In this study, interpretable machine learning uses the Shapley Additive Explanations (SHAP) theory to explain the Gradient Tree Boosting regression model (GBRT). SHAP model uses game theory marginal effects theory to explain the traditional machine learning black box, which satisfies the efficiency, symmetry, dummy and additivity of traditional game theory Shapely values, in addition to local accuracy, missingness and consistency, which can provide interpretable machine learning models from both macroscopic and microscopic perspectives, and thus are widely used in studies such as decoupling complex relationships [24,25]. The SHAP attribution value of the SHAP explanation tree model, i.e., the Shapely value calculation method, can be briefly expressed as [23]:

SHAP interprets the predicted value of the model as the sum of the imputed values of each input feature, i.e.,

$$g(z') = \varnothing_0 + \sum_{j=1}^{M} \varnothing_j z_j' \tag{2}$$

Since the input data of the tree model must be structured data, the above equation can be simplified, for instance, $x$ as:

$$g(x') = \varnothing_0 + \sum_{j=1}^{M} \varnothing_j \tag{3}$$

In the above equation, $g$ is the explanatory model, $z' \in \{0,1\}^M$ indicates whether the corresponding feature can be observed, $M$ is the special number, $\varnothing_j \in R$ is the attributed value of the $j$ feature, i.e., the Shapely value, and $\varnothing_0$ is the explanatory model constant.

When the model is nonlinear, or the features are not independent, then the SHAP imputation value is calculated according to the following equation:

$$\varnothing_j = \sum_{S \subseteq \{x_1, \cdots, x_p\} \setminus \{x_j\}} \frac{|S|!(p - |S| - 1)}{p!} \left( f_x(S \cup \{x_j\}) - f_x(S) \right) \tag{4}$$

where $\{x_1, \cdots, x_p\}$ is the set of all features, $P$ is the number of features, $\{x_1, \cdots, x_p\} \setminus \{x_j\}$ is the set of all features excluding $\{x_j\}$, and $f_x(S)$ is the predicted value of the feature subset $S$.

Gradient Tree Boosting is a generalization of Boosting integration based on arbitrary differentiable loss functions and can be used for regression in various domains, and under certain circumstances, GBRT outperforms Random Forests regression based on Bagging integration [34–36]. In this study, we used AFD, ALD, ADD, and TMP as features and AT as target variables and judged the influence factor contributions using Shapely scores. The hyperparameters of the GBRT were determined using a 10-fold cross-validation exhaustive grid method. The results of the hyperparameters are presented in the Table of GBRT hyperparameters (Table S2). The GBRT calculation can be briefly expressed as [36–38]:

Given $x_i$ is the predicted value of $y_i$ for $x_i$ is:

$$\hat{y}_i = F_M(x_i) = \sum_{m=1}^{M} h_m(x_i) \tag{5}$$

Constructed in greedy mode, we have:

$$F_M(x) = F_{M-1}(x) + h_M(x) \tag{6}$$

Given $F_{M-1}$, then:

$$h_M(x) = \operatorname*{argmin}_h L_m = \operatorname*{argmin}_h \sum_{i=1}^{n} l(y_i, F_{M-1}(x_i) + h(x_i)) \tag{7}$$

Using the first-order Taylor approximation, the value of the loss function $l$ can be approximated as:

$$l(y_i, F_{M-1}(x_i) + h(x_i)) \approx l(y_i, F_{M-1}(x_i)) + h_M(x_i) \left[ \frac{\partial l(y_i, F(x_i))}{\partial F(x_i)} \right]_{F=F_{m-1}} \tag{8}$$

Dividing the constant term gives:

$$h_M(x) \approx \operatorname*{argmin}_h \sum_{i=1}^{n} h(x_i) g_i \tag{9}$$

where $h_m$ is the weak learner. Gradient Tree Boosting uses decision tree regressors of fixed size as weak learners, $M$ is the number of base learners, and $(g_i)$ is $\left[ \frac{\partial l(y_i, F(x_i))}{\partial F(x_i)} \right]_{F=F_{m-1}}$.

For the historical data, least squares regression was used to analyze the characteristics of temporal trend changes in different agricultural zones. For the future scenario data, given that the current mainstream for temperature gridded data is based on bilinear interpolation [19], but bilinear interpolation does not consider the effect of elevation on temperature when interpolating temperature, so this method has a large error on the results of temperature interpolation. Therefore, in this study, the future scenario data are DEM-corrected for temperature using the temperature decrement rate, i.e., the observed temperature is converted to the temperature $T_0$ at sea level using the vertical decrement of temperature first, and then the DEM-corrected true temperature T-value is obtained using $T_0$ and $\delta H$ to interpolate to the same target accuracy grid as the historical data. The final MK abrupt change test is used to study the temporal variation characteristics in future scenarios. The above interpolation method can be expressed as [20]:

$$T = T_0 - \delta H \times 0.6 \times 0.01 \tag{10}$$

## 3. Results

### 3.1. Trend Analysis of AT Change in China from 1979 to 2018

3.1.1. Changes in the Spatial Distribution of AT

The average accumulation temperature of 1980–1989, 1990–1999, 2000–2009 and 2010–2018 were calculated, respectively, and the spatial distribution of accumulation temperature was obtained (Figure 2). The cold temperate zone is mainly concentrated in QTP, the Tianshan Mountains in the west of NAS, and the Daxing'an mountains in its east. The middle temperate zone is mainly concentrated in NEP, LP, the central Inner Mongolia Plateau in NAS, and the Qaidam Basin in the QT. The warm temperate zone is mainly concentrated in the northern part of 3HP, the Tarim Basin and Junggar Basin in the western part of NAS; the subtropical zone is mainly distributed in the southern part of 3HP, MYP, YGP, and the Sichuan Basin in SBS. The tropics are mainly distributed in the southern part of SC, i.e., Hainan Island and other areas. In addition, by comparing the four AT distribution maps in Figure 2, it can be seen that the AT shows a northward trend and retracts to higher elevations. The most significant northward trend is in the subtropical zone, which shows a significant northward trend in 3HP, while the tropical zone in SC, the southern part of QTP and the warm temperate zone in the northeastern plain area also show a significant northward trend. The trend of retraction to higher elevations is most significant in the warm temperate zone, which is in the Tarim Basin in the western part of NAS, while the cold temperate zone is in the Daxing'an mountains in the eastern part of

NAS, the northern part of YGP, the Hengduan Mountains in the western part of SBS, and the middle temperate zone is in LP and the central part of NAS.

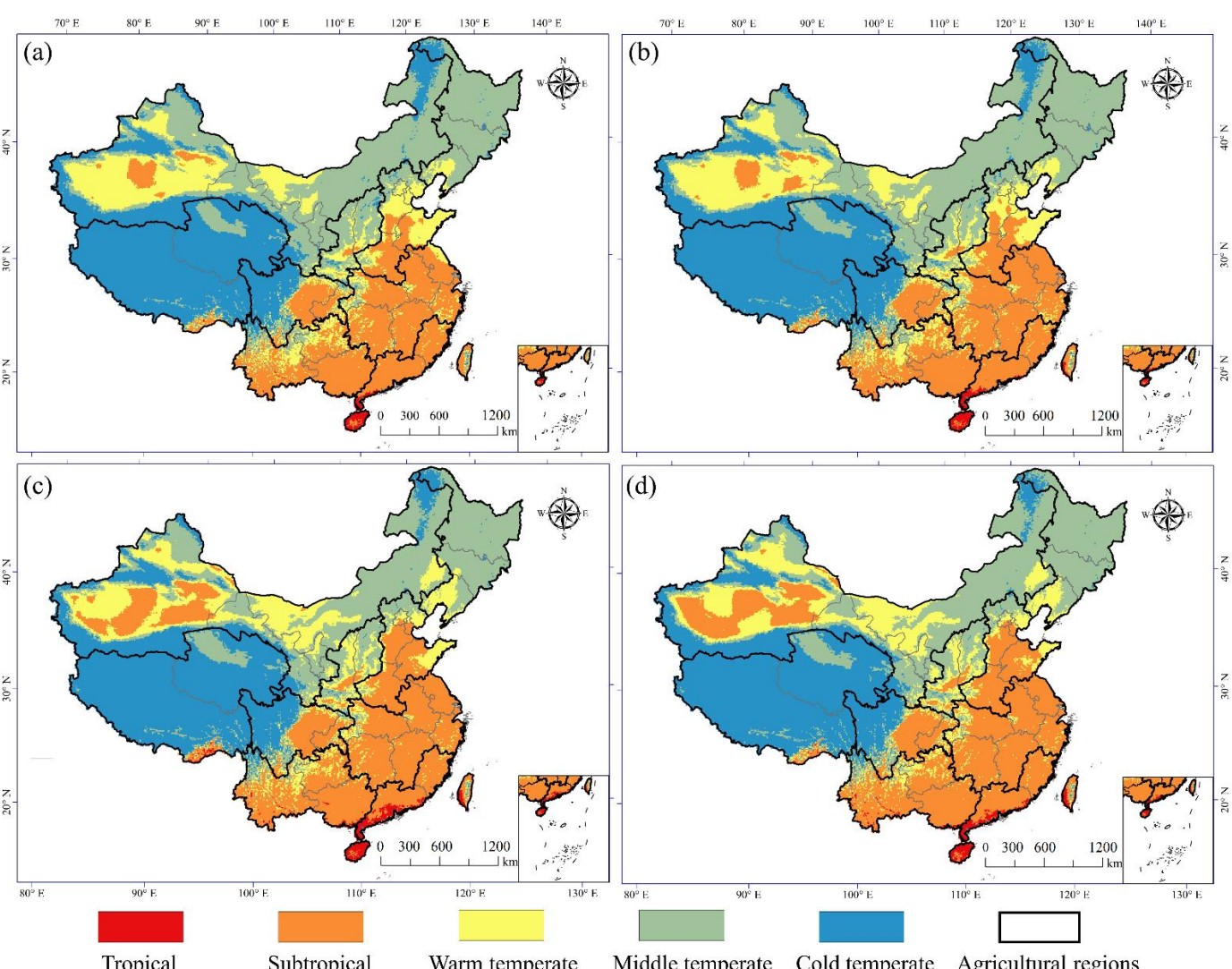

**Figure 2.** Spatial distribution of AT. Subplots (**a**–**d**) show the spatial distribution of mean AT for 1980–1989, 1990–1999, 2000–2009 and 2010–2018, respectively.

### 3.1.2. Spatial Variation in the Rate of AT Change from 1979–2018

The tendency rate of AT change was calculated using the MK trend test, and points that did not pass the 95% confidence level were excluded from plotting the trend of AT change from 1979–2018 (Figure 3). In the past four decades, the trend of AT in the vast majority of China has been significantly positive. Most of them have an AT change rate of more than 10 °C × d/a. Some areas have an AT change rate of more than 30 °C × d/a. The areas with an AT change rate of less than 0 °C × d/a and insignificant changes from 1979 to 2018 are mainly located in the eastern part of QTP along the Karakorum Mountains, some mountain ranges in the southern part of QTP, and the western part of NAS along the Pamir Plateau, Kunlun Mountains, and along the Alpine Mountains. The areas with a variation rate of 0–10 °C × d/a are mainly located in QTP, the Tarim Basin in the western part of NAS, the Daxing'an Mountains in the eastern part of NAS, and the Changbai Mountains in the eastern part of NEP. The areas where the rate of change of temperature is 10–20 °C × d/a are most widely distributed, mainly located in the central part of NAS, the western part of NEP, LP, 3HP, MYP, SC, and YGP and the eastern part of SBS. The areas

with a variation rate of 20–30 °C × d/a are mainly located in the eastern part of the SC region in Taiwan, the western part of YGP, the eastern part of MYP, the eastern part of SBS, the northeastern part of QTP in the Qaidam Basin, and the western part of NAS. The areas where the variation rate exceeds 30 °C × d/a are mainly located in the western part of NAS, the Taihang Mountains in the northeastern part of LP, the southern part of SBS, and the western part of YGP.

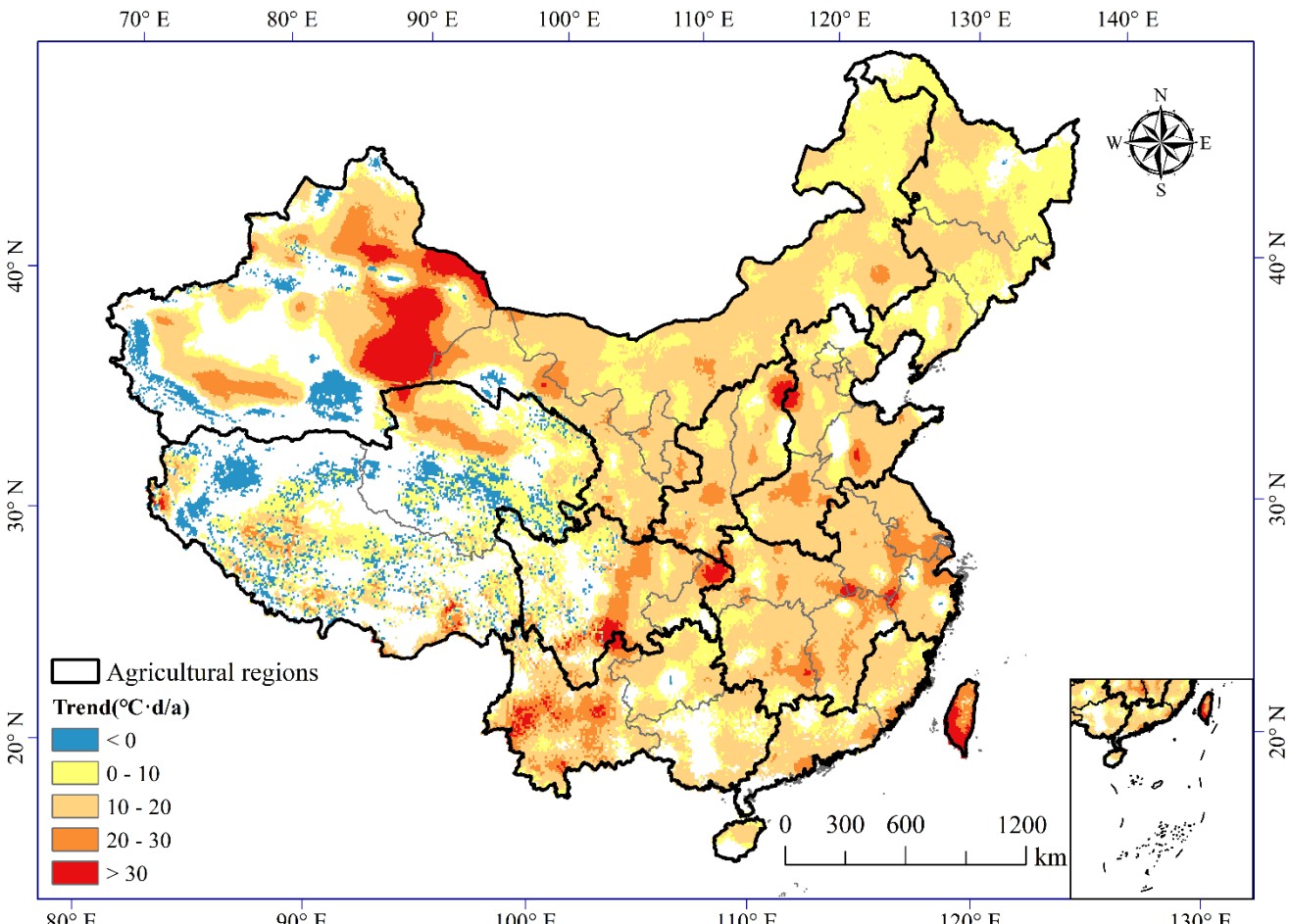

**Figure 3.** Trends in the AT from 1979–2018. The grid points that did not pass the significance test above the 95% confidence level were excluded.

### 3.1.3. Analysis of the Change in the Area of AT Belt

The number of grid points of each AT belt from 1979–2018 was counted, and the change in its area was analyzed (Figure 4). It can be seen that the area of the tropical and subtropical zones shows an increasing trend year by year, while the area of middle and cold temperate zones shows a decreasing trend year by year, while the area of the warm temperate zone shows a weak decreasing trend.

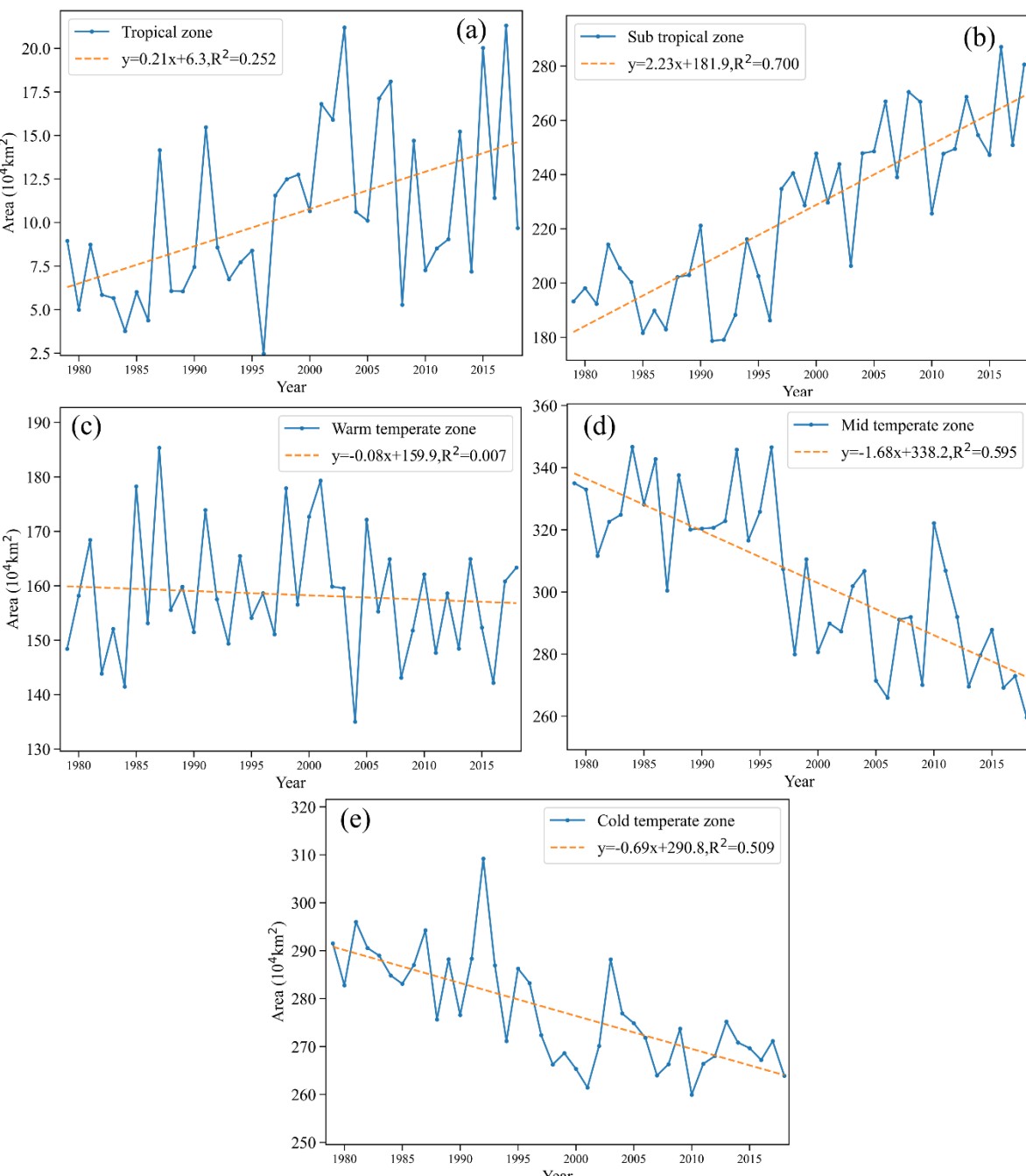

**Figure 4.** Variation in AT belt area from 1979 to 2018. Subplots (**a**–**e**) show the change in the area of AT belts from 1979 to 2018 in the tropical, subtropical, warm, middle, and cold temperate zones, respectively.

In detail, the subtropical area has increased significantly in the past four decades, with a change rate of up to $2.23 \times 10^4$ km$^2$/a, while the area of the middle temperate zone decreases significantly, with a rate of change of $-1.68 \times 10^4$ km$^2$/a. This indicates that with the northward shift of the AT belt, especially the northward expansion of the tropics and subtropics, the low AT belt area is continuously squeezed and eroded, resulting in the tightening of the area of the cold temperate zone as well as the middle temperate zone year by year. The northward expansion of the subtropics causes the most significant increase in area, and the middle temperate zone is most obviously affected by the northward shift of the AT belt.

The northward shift of the AT belt will lead to the change of crops in the middle of the cold temperate zones from mono-annual to tri-annual or even bi-annual, i.e., the change of crop maturity system. In addition, the northward shift of the AT belt will also cause potential changes in cropping structure, i.e., crops that could not be grown in the north due to the influence of temperature will appear in the original low AT belt region due to the increase in AT.

3.1.4. Analysis of the Temporal Variation of Regional AT and the Factors Influencing the AT

The trend of regional average AT changes in nine agricultural regions of China (Table 1) shows that the AT in nine agricultural regions of China showed an increasing trend from 1979 to 2018. This is consistent with the results derived from the spatial variation of AT (Figure 2). Among them, MYP had the fastest growth with a change rate of 15.739 °C × d/a. LP was the second with a change rate of 14.166 °C × d/a, while the QT had the slowest growth with a change rate of 4.687 °C × d/a. This is basically consistent with the results of spatial variation of AT.

**Table 1.** The trend of regional average AT changes from 1979 to 2018 in nine agricultural regions of China.

| Agricultural Regions | The regression Formula | $R^2$ |
|---|---|---|
| NEP | y = 11.533x − 20,380.439 | 0.721 |
| NAS | y = 8.877x − 15,118.768 | 0.540 |
| SC | y = 13.525x − 20,253.914 | 0.453 |
| 3HP | y = 12.950x − 21,683.497 | 0.613 |
| LP | y = 14.166x − 25,063.953 | 0.669 |
| QTP | y = 4.687x − 8904.137 | 0.702 |
| SBS | y = 11.816x − 20,844.801 | 0.714 |
| YGP | y = 13.180x − 21,046.067 | 0.602 |
| MYP | y = 15.739x − 26,263.058 | 0.677 |

From Figure 5, it can be seen that the contribution of each factor on AT changes in nine agricultural regions in China are different, and the main factor influencing the change of AT in all agricultural regions except LP and SC is the increase of ADD, while the main factor in LP is the advance of AFD, and the main factor in SC region is the increase of the TMP. In detail, the main influencing factor for the increase of AT in NEP, NAS, 3HP, MYP, SBS, and YGP is the increase in ADD, and the secondary influencing factor is the increase in TMP. The main influencing factor in LP is the advance of AFD, and the secondary influencing factor is the increase in ADD; in SC, the increase in TMP is the main reason for the increase of AT, and the increase in ADD is the secondary reason.

*3.2. Analysis of the Trend of China's AT Change in 2015–2100 under Different Shared Socioeconomic Pathways*

3.2.1. Comparative Analysis of the Spatial Distribution of Mean AT between 2090 and 2100 under Different Shared Socioeconomic Pathways

The spatial distribution of mean AT between 2090 and 2100 under different scenarios (Figure 6) and the spatial distribution of the average AT from 2010 to 2018 (Figure 2d) are compared. The tropical and subtropical zones will move further northward, and the cold and middle temperate zones will retract further to higher altitudes and higher latitudes.

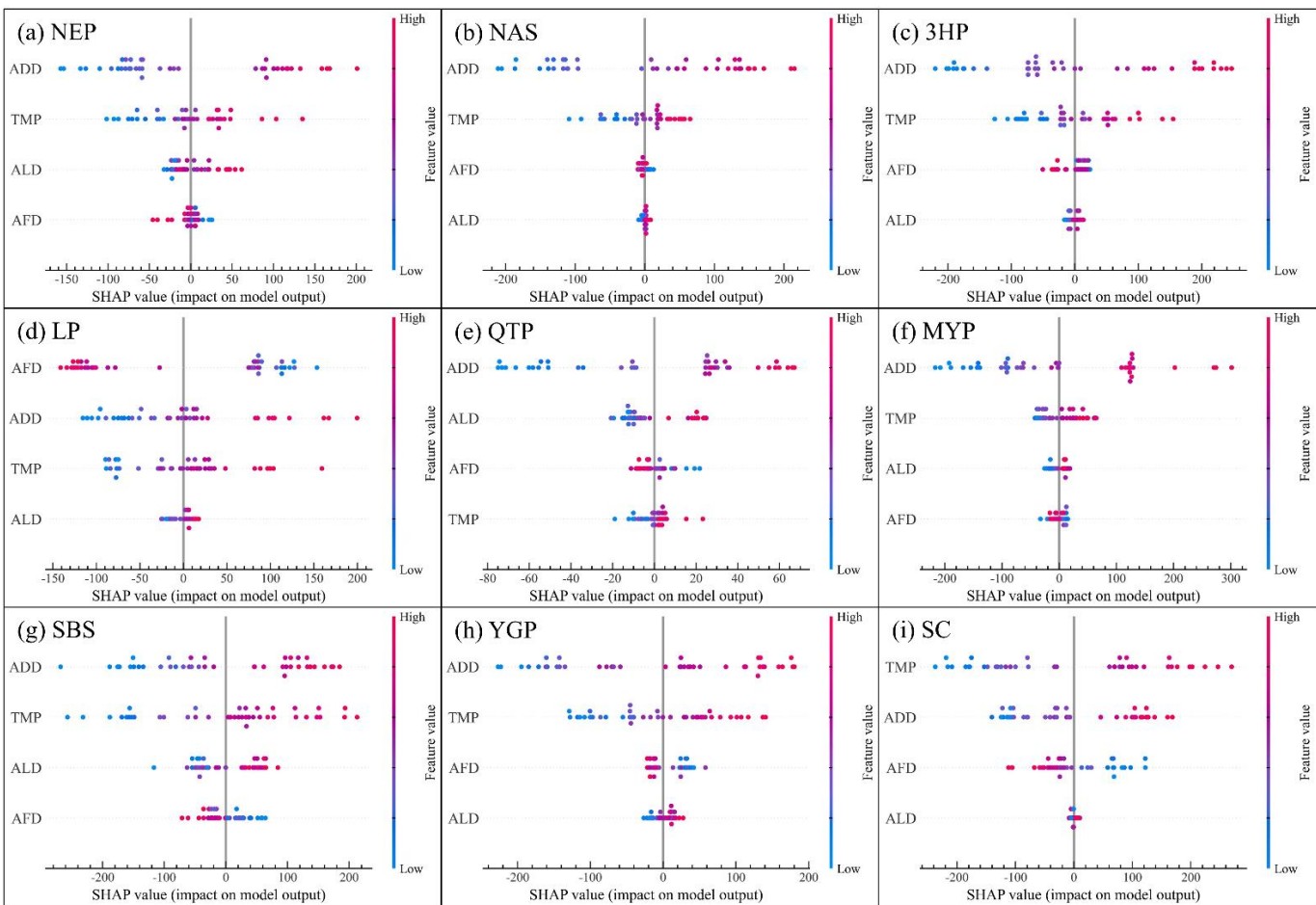

**Figure 5.** The contribution of each factor on AT changes from 1979 to 2018 in nine agricultural regions of China. Subplots (**a**–**i**) are the Shapely values in the agricultural area of NEP-SC, respectively.

Under the SSP1-2.6 scenario, the spatial distribution of mean AT between 2090 and 2100 remains largely unchanged compared to that of 2010–2018. Under the SSP2-4.5 scenario, the cold temperate zone in the eastern Daxing'an Mountains of NAS will basically cease to exist and will be completely replaced by the middle temperate zone; while the middle temperate zone in the central part of NAS and the junction of LP will be completely replaced by the warm temperate zone; the tropics will initially appear in the southern part of YGP, and the tropics in the southern part of SC will move further northward. The AT belt in other regions will remain basically unchanged. Under the SSP3-7.0 scenario, the cold temperate zone in the eastern part of NAS will be completely replaced by the middle temperate zone, and the middle temperate zone at its junction with LP will be completely replaced by the warm temperate zone and the subtropical zone, while the western part of the region will become completely subtropical except for the high-altitude region; the warm temperate zone in the southern part of NEP will move further northward and reach Heilongjiang Province. The tropical zone will shift further northward, which is more obvious than the SSP2-4.5 scenario. In the SSP5-8.5 scenario, the northward shift of the AT belt is most significant. In 2090s, the middle temperate zone will retract to the Daxing'an Mountains, located in the eastern part of NAS. The warm temperate zone will fully occupy the northern and central parts of NEP and the eastern part of NAS. The subtropical zone will fully cover the central and western parts of NAS, LP, the southern part of NEP, 3HP, MYP, YGP, and the eastern part of SBS. The SC will be completely covered by the tropics.

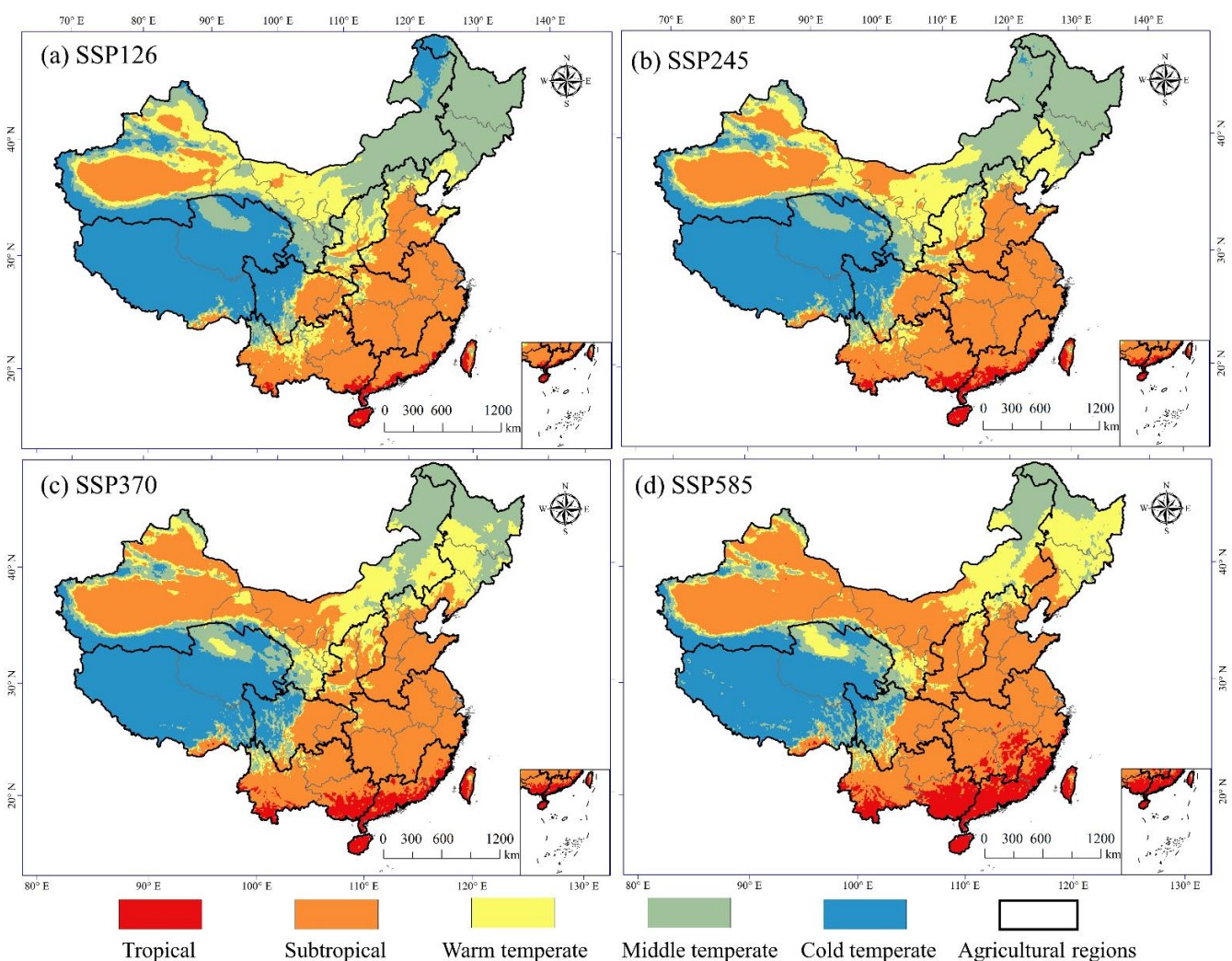

**Figure 6.** Spatial distribution of mean AT between 2090 and 2100 under the Shared Socioeconomic Pathway. Subplots (**a**–**d**) show the spatial distribution of mean AT for 2090–2100 under SSP126, SSP245, SSP370 and SSP585 scenarios, respectively.

3.2.2. Analysis of Spatial Trends of AT from 2015 to 2100 under Different Shared Socioeconomic Pathways

The majority of the regions in the country all show a significant increasing trend in the AT between 2015 and 2100 under different shared socioeconomic pathways, and the regions with a decreasing trend in AT are mainly located in QTP. The growth trend in the northern agricultural areas is not as rapid as that in the southern agricultural areas. (Figure 7) The areas with the fast-increasing trend of accumulation temperature are mainly located in MYP, the eastern part of SBS, YGP, and SC. The regions with lower trends of accumulation temperature growth are mainly located in NEP and QTP. In addition, under the SSP1-2.6 scenario, the 2015–2100 AT trends in NAS, QTP, and NEP are not significant (Figure 7a).

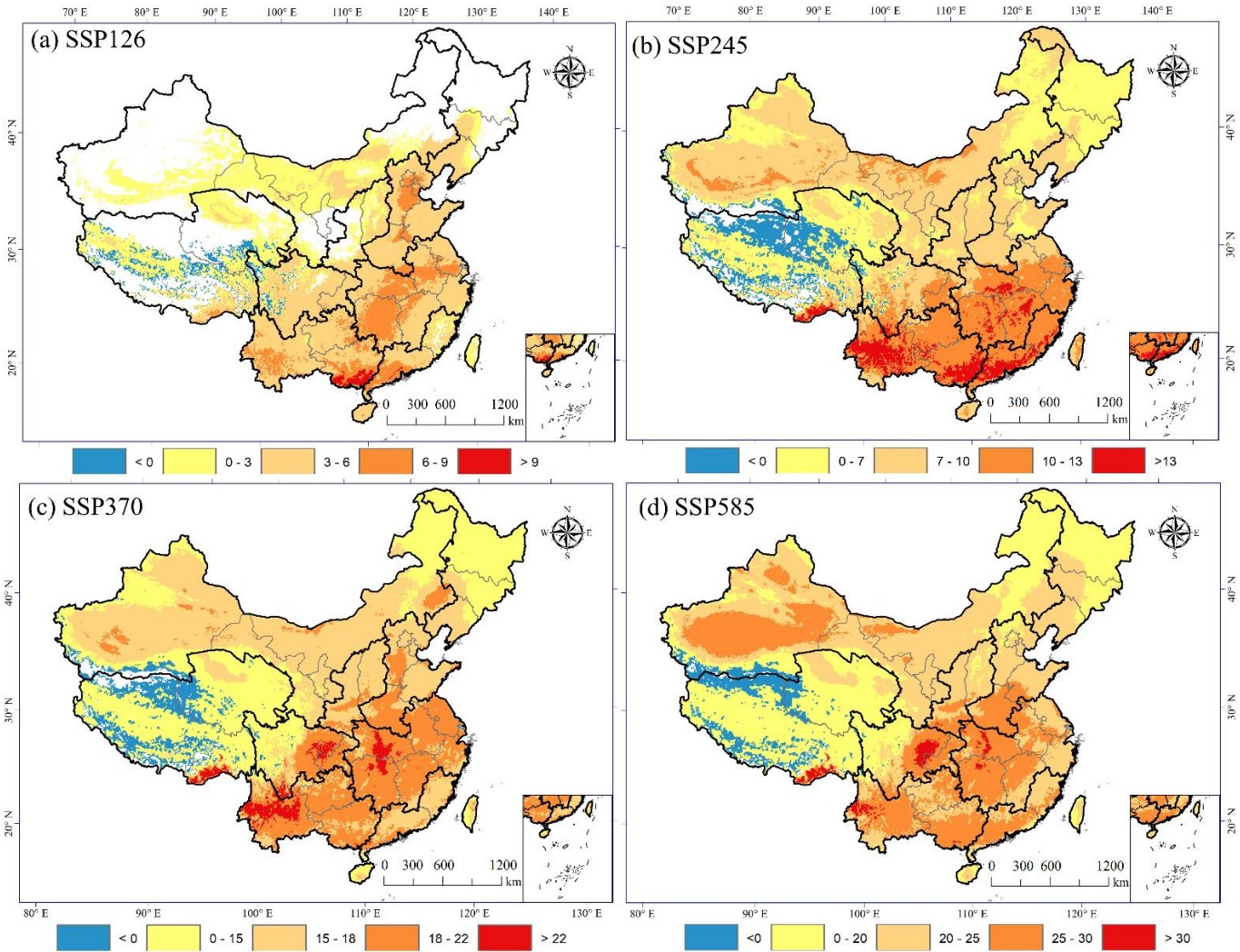

**Figure 7.** Trends in the AT between 2015 and 2100 simulated by the model under the Shared Socioeconomic Pathway. Subplots (**a–d**) show the spatial distribution of the 2015–2100 rate of change obtained using the MK trend test for the SSP126, SSP245, SSP370, and SSP585 scenarios, respectively. The grid points that did not pass the significance test above the 95% confidence level were excluded.

### 3.2.3. Analysis of AT Belt Area Change from 2015 to 2100 under Different Shared Socioeconomic Pathways

The area of the AT belt from 2015 to 2100 under the four scenarios was counted to analyze the rate of change of the AT belt area (Figure 8). The area of the tropical and subtropical zones continues to grow, while the area of the middle temperate zone and cold temperate zones continues to shrink, and the area of the warm temperate zone remains basically unchanged. The growth of the subtropical zone is the most significant, and its growth rate of change can reach $2.55 \times 10^4$ km$^2$/a under the SSP5-8.5 scenario; the shrinkage of the middle temperate zone is the most significant, and its decrease rate of change can reach $-1.93 \times 10^4$ km$^2$/a under the SSP5-8.5 scenario. This indicates that under the future scenario, the AT belt will shift further northward, the area of the high AT belt in the tropics and subtropics will grow further, and the area of the low AT belt in the middle temperate zone and cold temperate zones will be restricted by the impact, which will have a profound impact on the agricultural cropping structure and major crop types.

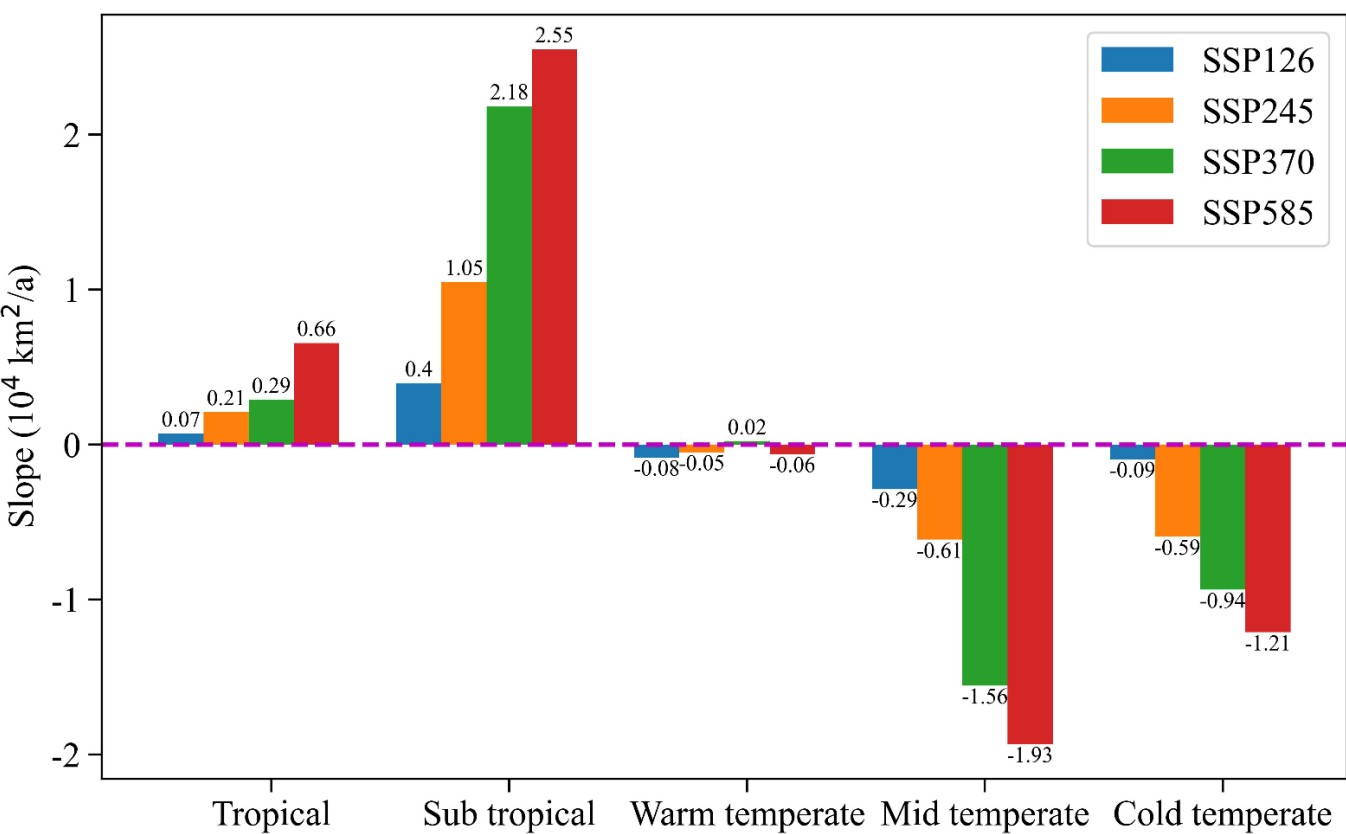

**Figure 8.** Area change rate of each AT belt from 2015 to 2100 under the Shared Socioeconomic Pathway.

In addition, compared to the historical scenario AT belt area change, the AT belt change under the SSP1-2.6 scenario, i.e., the low forcing radiation-based sustainable development shared socioeconomic pathway, is negligible. The historical subtropical area change rate can reach $2.23 \times 10^4$ km$^2$/a, while the subtropical area change rate in the SSP1-2.6 scenario is only $0.4 \times 10^4$ km$^2$/a; the historical area change rate of the middle temperate zone is $-1.68 \times 10^4$ km$^2$/a, while the change rate in SSP1-2.6 scenario is only $-0.29 \times 10^4$ km$^2$/a.

3.2.4. Analysis of Temporal Variation of AT and Influence Factors under Different Shared Socioeconomic Pathways

The mean AT of 2015–2100 simulated by the model under different scenarios was calculated, and the MK abrupt change test was used to detect the mutation and plot the MK abrupt change curve (Figure 9). From the MK abrupt change curve, it can be seen that there is only one abrupt change in the AT under different scenarios, and all of them show an increasing trend. All scenarios show a fluctuating decreasing trend in the first few years and basically show a significant increasing trend after 2030. The SSP1-2.6 scenario has the largest fluctuation, while the SSP3-7.0 and SSP5-8.5 scenarios have the most stable increase.

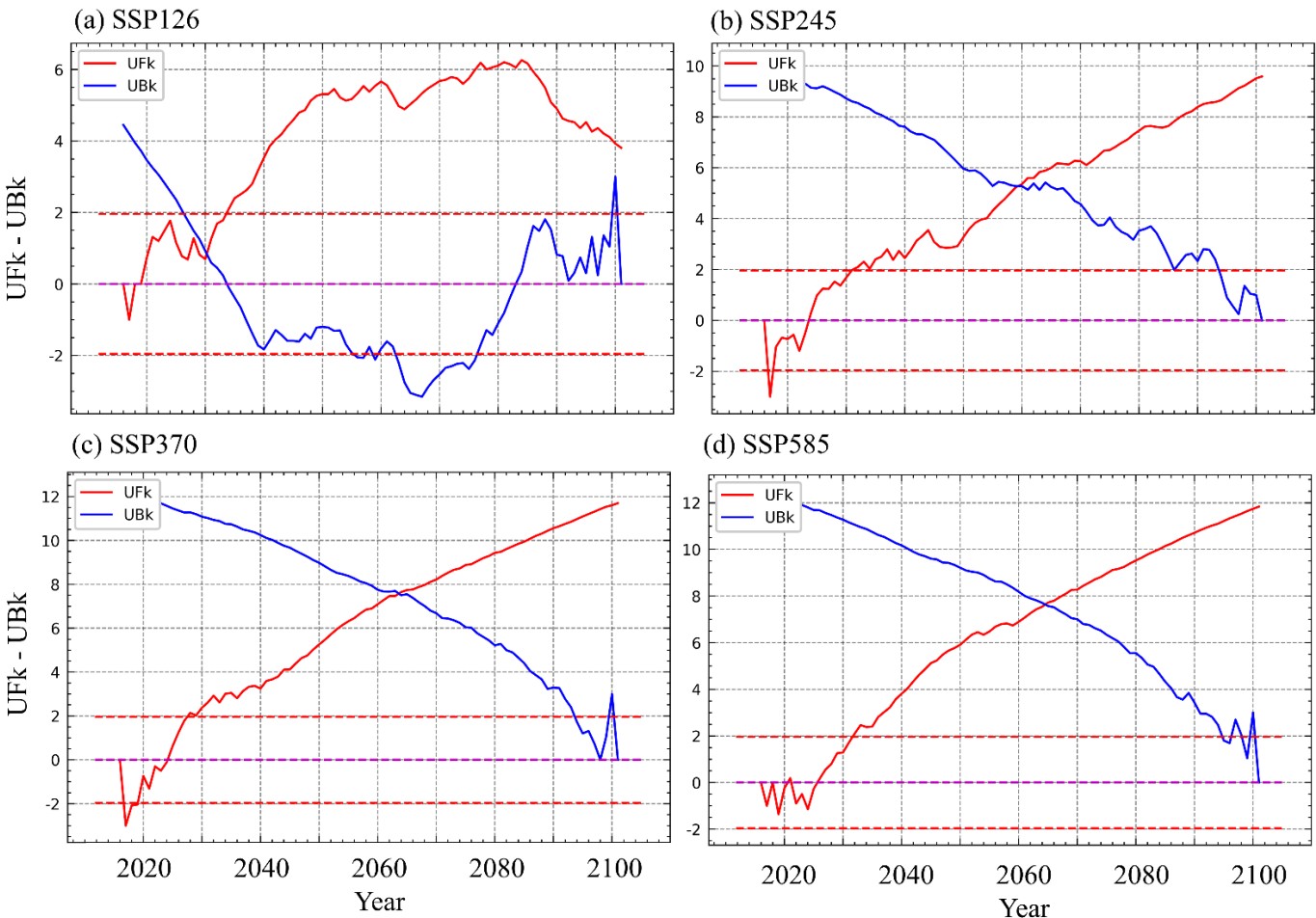

**Figure 9.** MK abrupt change curve for different the Shared Socioeconomic Pathways of AT change. Subplots (**a**–**d**) are distributed as MK abrupt change curves for 2015–2100 under SSP126, SSP245, SSP370 and SSP585 scenarios. The red line in the graph represents the UFk value and the purple line represents the UBk value.

The SSP1-2.6 scenario shows an abrupt change in AT around 2030, a decreasing trend until 2020, and an increasing trend in AT from 2020 to 2035, with large fluctuations during the period and a significant increasing trend after 2035. The abrupt change in AT under the SSP2-4.5 scenario occurs around 2060, the abrupt change in AT under the SSP3-7.0 scenario occurs around 2065, and the sudden change in AT under the SSP5-8.5 scenario occurs around 2065. The trends of the three scenarios, except SSP1-2.6 are generally consistent, with a decreasing trend in AT until 2025, a stable increasing trend between 2025 and 2030, and a stable, increasing trend after 2030.

By comparing the SHAP scores from 2015 to 2100 under the four different shared socioeconomic pathways (Figure 10), we obtained the main factor influencing the change in AT under different SSPs. In the medium and low radiative forcing scenarios, the main factor influencing the change in AT is the increase in ADD, followed by the increase in TMP (Figure 10a,b), while in the high radiative forcing scenario, the main factor influencing the change in AT is the increase in ADD, followed by the delay in ALD (Figure 10c,d).

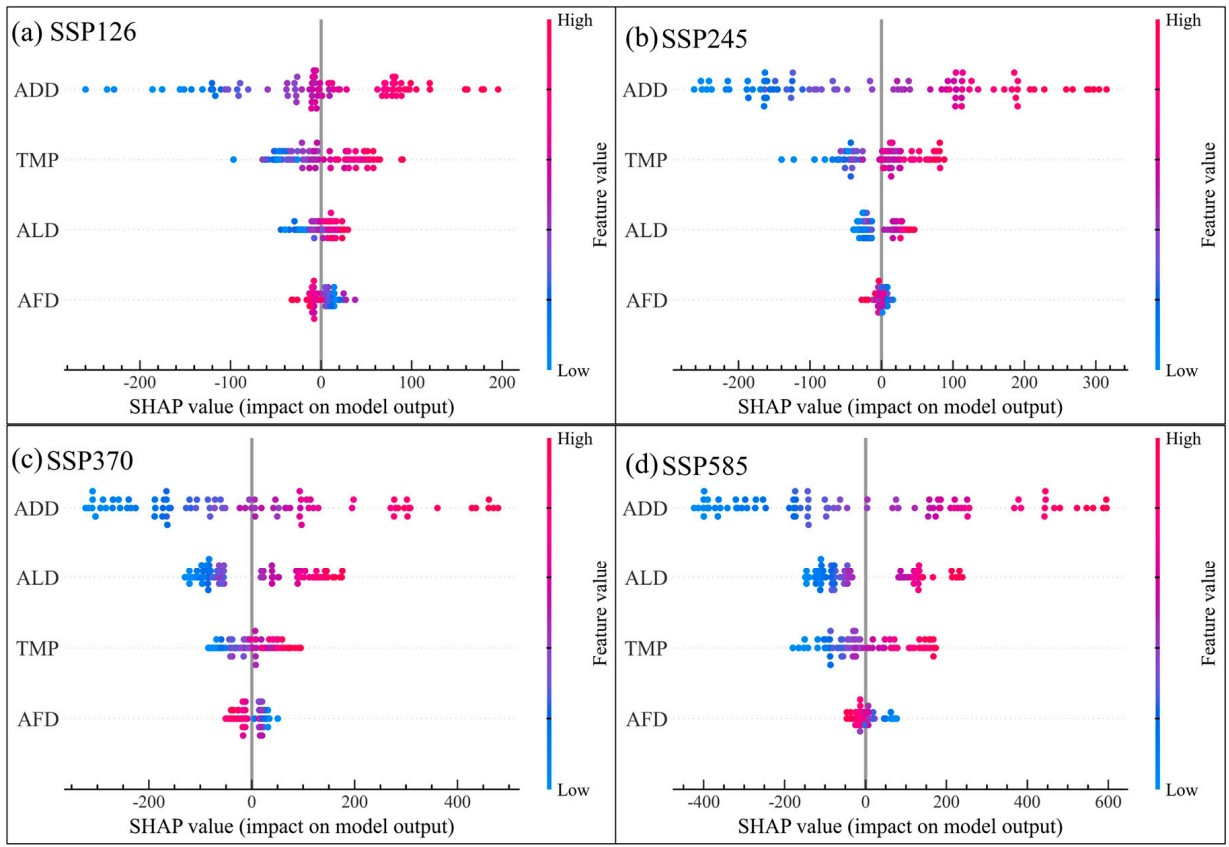

**Figure 10.** Influence factors of AT for 2015-2100 under the Shared Socioeconomic Pathway. Subplots (**a**–**d**) shows the Shapely values for SSP126, SSP245, SSP370 and SSP585 scenarios.

By comparing the SHAP scores of nine agricultural regions in China for 2015–2100 under four different scenarios (Figure 11), it can be seen that in SC, except for the SSP1-2.6 scenario where the main influencing factor was the increase in ADD, all other scenarios had the increase in TMP as the main factor influencing the increase in AT, with the higher forcing type being the most obvious. Except for the SSP5-8.5 scenario, all other scenarios in YGP were mainly influenced by the increase in ADD. In NEP, NAS, LP, QTP, MYP, and SBS, the main influencing factor is the increase in ADD, and the secondary influencing factor is the increase in TMP; in 3HP, the main and secondary influencing factors of AT show different characteristics with the change of forcing types and social sharing paths, among which SSP1-2.6 and SSP3-7.0 scenarios show a balanced trend, i.e., the effects of ADD and TMP increase remain the same, while the effect of TMP increase on AT increase is much greater than that of ADD in SSP2-4.5 scenario, and SSP5-8.5 is just the opposite of SSP2-4.5 scenario, where the effect of ADD increase on AT increase is greater than that of TMP increase.

Overall, the main influence of SSP1-2.6 is the increase in ADD in all agricultural regions, and the increase in TMP is the secondary cause of AT change in all agricultural regions; the increase in TMP and ADD in all agricultural regions in SSP2-4.5 is the main cause of AT change, and ADD is the main cause; in the SSP3-7.0 scenario, except for SC, where the increase in TMP is the main factor influencing the change in AT, the increase in ADD is the main factor influencing all other agricultural regions; in the SSP5-8.5 scenario, except for YGP and SC, where the increase in TMP is the main factor influencing the increase in AT.

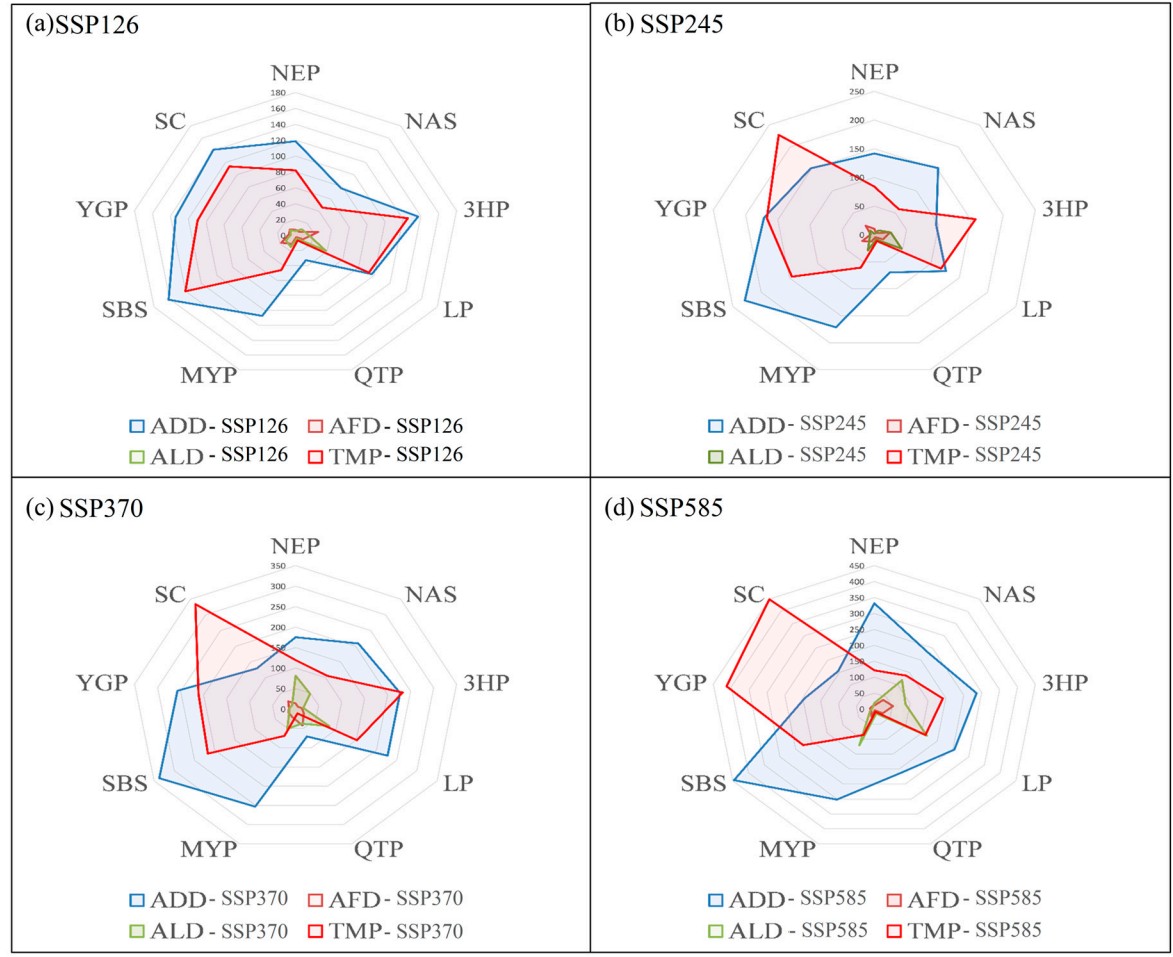

**Figure 11.** Main influencing factors of nine agricultural regions in China for 2015–2100 under the Shared Socioeconomic Pathway. Subplots (**a–d**) are Shapely values in different agricultural zones under SSP126, SSP245, SSP370 and SSP585 scenarios.

## 4. Discussion

### 4.1. Contributors for the Changes in AT

Changes in AT are mainly influenced by changes in ADD, and the increase in ADD from 1979 to 2018 is more significantly influenced by the advancement of AFD compared to the delay in ALD (Figure 4), while the future scenario shows the opposite result (Figures 10 and 11). ZHAO H et al. showed by Pearson correlation coefficient analysis that ADD in the Republican basin from 1953 to 2010 was significantly and negatively correlated with AFD [12]. LI S et al. showed that AFD, ALD, and ADD were closely related to vegetation phenology [9], while Huanjiong Wang et al. showed that cold excitation increased the AT demand of plants in spring and had a delaying effect on spring phenology [39]. HOU P et al. showed that the coefficient of variation between maize seeding emergence and AT was greater in northern China and major plains than in other growing periods [40], which echoed the early change in ADD; Shen, M. G et al. showed that the vegetation start of the season (SOS) was advanced in the Tibetan Plateau while the vegetation end of the season (EOS) was delayed, and the EOS delay was more pronounced under the Shared Socioeconomic Pathway 5-8.5 [41], which was consistent with the delay in ALD changes revealed from our study. At the present stage, crop phenology is more focused on the early flowering period and sowing period of crops [39,42], while for the future scenario, the effect of delayed ALD is greater than that of AFD; therefore, delayed vegetation phenology should receive more attention.

*4.2. Northern Shift of the AT Belt and Change in the Northern Boundary of Planting*

The AT belt mainly shows a northward shift and retraction trend toward higher elevations (Figure 2), and the middle and cold temperate zones are tightened by the northward expansion of the tropics and subtropics year by year (Figure 3), and the trend continues to be maintained under future scenarios (Figures 6 and 8). Kong F et al. showed that the AT belt in China showed a northward and westward expansion from 1961 to 2018, and the high AT belt expanded toward higher elevations [8]. BAI L et al. showed that the increase in the area of tropical and subtropical regions in mainland China from 1961 to 2018 caused a decrease in the area of cold and middle temperate zones, which would lead to a northward shift in the cultivation boundary of tropical and subtropical crops [10]. CHEN X et al. pointed out that the high and stable yield of winter wheat in northern China was characterized by a decrease from northeast to southwest [43], which is consistent with the trend of northward shift of the AT belt in China. It is worth noting that the sustainable social shared path scenario will effectively reduce carbon emissions compared with other scenarios, which will greatly mitigate and curb the northward shift of the AT belt (Figures 6a and 8a) and has positive implications for the maintenance and stabilization of cropping structure and crop maturity changes.

*4.3. Recommendations*

Since the spatial and temporal variabilities of AT are more stable under the sustainable development path of the future scenario than under other scenarios (Figures 6 and 8), this is of great importance for the stability of cropping systems. Therefore, policymakers should promote and support sustainable development, reduce radiative forcing, and reduce emissions. Agricultural area planners, on the other hand, should take into account the fact that AT can affect cropping systems and phenological changes and should consider the current situation, turn crises into opportunities, and make good agricultural management plans according to local conditions to avoid the adverse effects caused by the northward shift of the AT belt.

*4.4. Shortcomings and Prospects of This Study*

In this paper, the statistical downscaling method based on topographic correction was used to downscale future scenario data, and the use of the dynamic downscaling method for spatial interpolation should be considered in the subsequent study. Meanwhile, due to the limitation of available data, only the AT trend from 1979 to 2018 can be studied. In the follow-up study, the grid data from 1960 to 2020 should be selected to improve the time breadth of the study so as to better study the historical AT trend in China. In addition, since there are various factors influencing AT, only TMP, ADD, AFD and ALD are considered as influencing factors in this paper, and it is found in the study that AT and elevation still have a certain relationship. Therefore, the relationship between DEM, latitude and longitude, etc., and AT should be considered in the subsequent study. Finally, the national grain production data were not available for the whole period from 1979 to 2018, which hindered us from further analyzing the relationship between AT and grain yield for this long period. Therefore, in our future works, we will further investigate the relationship between grain yield and its meteorological factors (such as AT, air temperature, precipitation, solar radiation, etc.) based on this study.

**5. Conclusions**

In this paper, the spatial and temporal evolution of AT in nine agricultural regions of China was obtained by MK abrupt change and trend test, and the area change trend of each AT zone was obtained by using the least squares analysis. The dominant factors of AT in the nine agricultural regions were obtained by comparative analysis using interpretable machine learning models. The main conclusions are as follows.

1.    The AT in China from 1979 to 2018 mainly shows a trend of northward shift and retreat to higher elevations. The most significant northward trend is in the subtropics, and

the trend of retreating to higher altitudes is most significant in the warm temperate zone. In 2090–2100, the trend of northward shift and retraction to higher altitudes of the AT belt remains unchanged;

2. In the past forty years, with the northward shift of the AT belt, especially the northward expansion of the tropics and subtropics, the low AT belt has been continuously squeezed and eroded, resulting in the narrowing of the cold temperate zone and the middle temperate zone year by year. Among them, the area increase caused by the northward expansion of the subtropics is the most significant, and the middle temperate zone is most obviously affected by the northward shift of the AT belt. In the future scenario, the development pattern of the area of the AT belt remains basically the same, i.e., the high AT belt will continue to expand northward and continuously squeeze and erode the area of the low AT belt;

3. Except for LP and SC, the main factor affecting the change of AT in 1979–2018 is the increase of ADD in all other agricultural regions, the main factor in LP is the advance of AFD, and the main factor in SC is the increase of TMP. In the future scenario, the influence of TMP on the nine agricultural regions increases sequentially from the lower radiative forcing type to the higher radiative forcing type and ADD is always the main influencing factor of the AT change in the nine agricultural regions. In addition, the contribution of the advance of AFD is larger than that of ALD in 1979–2018, while the opposite is true in the future scenario.

**Supplementary Materials:** The following supporting information can be downloaded at: https://www.mdpi.com/article/10.3390/agronomy13051203/s1, Table S1: The CMIP prediction and historical evaluation; Table S2: Table of GBRT hyperparameters.

**Author Contributions:** Methodology, formal analysis, visualization, software and writing—original draft preparation X.L.; conceptualization, supervision, writing—review and editing, and funding acquisition, L.Y.; resources, investigation and project administration, Q.Y. and L.B.; validation, data curation, G.L., J.Y., X.C. and X.G. All authors have read and agreed to the published version of the manuscript.

**Funding:** This study was supported by the Strategic Priority Research Program (A) of the Chinese Academy of Sciences (XDA28080503), the National Natural Science Foundation of China (42071025) and the Youth Innovation Promotion Association of Chinese Academy of Sciences (2023240).

**Data Availability Statement:** The original contributions presented in the study are included in the article/supplementary material; further inquiries can be directed to the corresponding author.

**Acknowledgments:** The authors thank the National Tibet Plateau Scientific Data Center, the Resource and Environment Science and Data Center as well as CMIP6 for data support. The historical temperature data is provided by National Tibetan Plateau Data Center (http://data.tpdc.ac.cn (accessed on 3 July 2022)). The future scenario data is provided by the ScenarioMIP dataset (https://esgf-node.llnl.gov/search/cmip6 (accessed on 29 July 2022)). The data of China's nine major agricultural divisions and provincial administrative boundaries is provided by the Resource and Environment Science and Data Center (https://www.resdc.cn/Default.aspx (accessed on 19 June 2022)).

**Conflicts of Interest:** The authors declare no conflict of interest.

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
