# Peer review of "Temporal Trends and Future Projections of Accumulated Temperature Changes in China"

_agronomy, doi:10.3390/agronomy13051203_

Round 1
Reviewer 1 Report
Temporal Trends and Future Projections of Accumulated Temperature Changes in China
The authors do an excellent job of researching the historic and projected changes in the climate variables. The article is well written and the quality of the graphics is high. The research is highly relevant given the implications for the future of agriculture in the country.
Abstract
The final sentence is very long and the final phrase “on the contrary, the changes of vegetation phenology delay should pay more attention” is an incomplete thought. I assume the authors are suggesting that the delay requires that we pay more attention, or that that trend is an important one to track.
The study found that (1) the accumulated temperature belt shows a northward shift and retreat trend toward higher altitudes, and the area of the high accumulated temperature belt increases year by year, leading to the narrowing of the area of the low accumulated temperature belt year by year, and the trend remains unchanged under the future scenario; meanwhile, the northward shift trend of the accumulated temperature belt is greatly mitigated and curbed under the SSP126 scenario; (2) the changes of accumulated temperature belt the main influencing factor is the increase of accumulated temperature duration days, and the secondary factor is caused by the increase of temperature, and the contribution brought by the first day of accumulated temperature from 1979 to 2018 is greater than that brought by the last day of accumulated temperature, while in the future scenario, on the contrary, the changes of vegetation phenology delay should pay more attention.
Introduction
Line 43
Suggest changing the bears the brunt of the response …
The accumulated temperature (AT), as the sum of the average daily temperature in a region over a period of time, bears the brunt of the response to the increase in global average temperature.
Line 47
Suggest changing with different growing areas …. to in different growing areas
while playing a key role in the cropping system transition with different growing areas [8].
Line 71
This sentence is unnecessary it could be deleted or changed to specify how the studies in this paragraph reflect a different analyses than the previous paragraph.
There are more research methods on the analysis of AT variation and its dominant factors.
Line 80 – is delicate a standard term in the field or is the implication that this method is more precise, more sensitive, or likely to be easily biased?
It has been shown that the spatial characteristics obtained by interpolation extrapolation using grid point data are more delicate than those obtained by interpolation of station data [10];
Line 121. There are nine agricultural regions – and labels on the map indicate A-I. The Figure caption says A-H.
Figure 1. Nine major agricultural regions in China and the distribution of meteorological stations. The A-H are NEP, NAS, 3HP, LP, QTP, MYP, SBS, YGP and SC. The red triangle is the national meteorological station.
Author Response
Dear Reviewer,
We would like to express our sincere gratitude for your thorough review and valuable feedback on our research. Your suggestions and criticisms have greatly contributed to improving the quality of our work. We appreciate your time and effort in providing detailed comments and constructive criticism, which have helped us to refine our methodology and strengthen our results.

Reviewer 2 Report
Review of the article "Temporal Trends and Future Projections of Accumulated Temperature Changes in China”
The manuscript deals with a topic that is of interest in the areas of agricultural as well as atmospheric sciences applications, especially to the scientific community interested in assessing the impacts of climate change on agriculture. The authors focused on “analyzing the spatial and temporal variation of accumulated temperature in China from 1979 to 2018 based on the perspective of agricultural regions, and predicts the future variation based on CMIP6 to find the evolution pattern of its change characteristics and search for the differences of the main influencing factors among different agricultural regions”.
Although the manuscript deals with a topic relevant in the scope of studies about crop phenology and cropping system relevant for the improvement of grain production, it requires more detail and improvements, in order to explicit/complement and be suitable for publication.
General Comments
- The subject is appropriate to "Agronomy" and the paper contains some relevant results, especially related to temporal trends of accumulated temperature changes in a very big country, where food supply is crucial for the population and can be significantly impacted by climate change impacts.
- In general aspects, the paper is generally well-structured; however, it requires more detail and improvements in order to contribute for subsidize the climate change adaptation planning.
- Considering the extent of areas that may be impacted by higher temperature accumulations in future scenarios, it is relevant to include more detailed information regarding the decrease in grain production in the each agricultural region evaluated.
- Precipitation is one of the most significant climatic parameters and its variability and trends have great influences on environmental and grain production. Justify why precipitation data were not evaluated together with temperature?
- What are the main crops grown in each region? How many in situ stations are in China for monitoring rainfall and temperature in each agricultural region?
- Results and Discussions
This topic needs to be better detailed / discussed. The results are presented, highlighting the variability presented in the generated figures, without discussing how the variability found will impact grain yield in quantitative terms.
- Considering the evaluated period (1979-2018), what is the impact on grain production resulting from changes in the AT belt in quantitative terms?
- It would be essential/relevant to add a section/paragraphs discussing the strengths, as well as the limitations of the current study and how can support policymakers and the farming communities to design proactive coping strategies related to different scenarios of climate changes.
Author Response

(The authors gave the same response as above.)

Round 2
Reviewer 2 Report
The authors included part of the suggested recommendations in the manuscript, as well as justified the limitations found to meet another part of the postulated suggestions.
I consider the article ready to be published, emphasizing that the limitations are clearly explained in the final version.